# Quasiparticle tunnel electroresistance in superconducting junctions

V. Rouco[1,5], R. El Hage[1], A. Sander [1], J. Grandal[2], K. Seurre [1], X. Palermo [1], J. Briatico [1], S. Collin[1], J. Trastoy[1], K. Bouzehouane [1], A.I. Buzdin [3], G. Singh [4], N. Bergeal [4], C. Feuillet-Palma[4], J. Lesueur [4], C. Leon [2], M. Varela [2], J. Santamaría[1,2] & Javier E. Villegas [1*]

The term tunnel electroresistance (TER) denotes a fast, non-volatile, reversible resistance switching triggered by voltage pulses in ferroelectric tunnel junctions. It is explained by subtle mechanisms connected to the voltage-induced reversal of the ferroelectric polarization. Here we demonstrate that effects functionally indistinguishable from the TER can be produced in a simpler junction scheme—a direct contact between a metal and an oxide—through a different mechanism: a reversible redox reaction that modifies the oxide's ground-state. This is shown in junctions based on a cuprate superconductor, whose ground-state is sensitive to the oxygen stoichiometry and can be tracked in operando via changes in the conductance spectra. Furthermore, we find that electrochemistry is the governing mechanism even if a ferroelectric is placed between the metal and the oxide. Finally, we extend the concept of electroresistance to the tunnelling of superconducting quasiparticles, for which the switching effects are much stronger than for normal electrons. Besides providing crucial understanding, our results provide a basis for non-volatile Josephson memory devices.

[1] Unité Mixte de Physique, CNRS, Thales, Université Paris-Saclay, 91767 Palaiseau, France. [2] Grupo de Física de Materiales Complejos, Dpto. Física de Materiales, Universidad Complutense de Madrid, 28040 Madrid, Spain. [3] Université de Bordeaux, LOMA UMR CNRS 5798, F-33405 Talence, France. [4] Laboratoire de Physique et d'Etude des Matériaux, ESPCI Paris, Université PSL, CNRS, 75005 Paris, France. [5] Present address: Grupo de Física de Materiales Complejos, Dpto. Física de Materiales, Universidad Complutense de Madrid, 28040 Madrid, Spain. *email: javier.villegas@cnrs-thales.fr

Experimental realisations of the tunnel electroresistance[1–5] (TER) have appeared during the last decade in ferroelectric tunnel junctions, which consist of an ultrathin (few nm) ferroelectric tunnel barrier sandwiched between two dissimilar electrodes[3,5] (see Fig. 1a for a sketch). In most of the existing experiments, at least one of the involved materials (ferroelectric and/or electrodes) is a complex oxide. These junctions show a characteristic switching between two (or more) non-volatile resistance states that is obtained by applying few-volts pulses across the ferroelectric barrier. Two key functional properties make TER fundamentally different (and technologically interesting) as compared to other resistance-switching phenomena in oxides[6]. First, the conduction mechanism (electron tunnelling) yields sizable current densities at low bias (~mV), which facilitates the non-destructive readout of the resistance states. Second, the resistance switching may be accompanied by a change of the electrode's physical properties[7–10], which enriches the related

physics and potential applications. These include memories[11] and memristors[12] for neuromorphic computing[13,14].

A series of mechanisms explain the TER under the premise that the applied voltage pulses reverse the ferroelectric polarisation[2,15,16]: changes of the orbital hybridisation at the junction interfaces (which modify the probability of electron transmission)[2], piezoelectric effects (which modify the tunnel barrier thickness)[17], and effects related to the screening of the polarisation charges[15]. The latter effects may operate at two levels. First, because dissimilar electrodes have different Thomas-Fermi screening length $\lambda_{TF}$, the height of the energy barrier across which electrons tunnel depends on whether the ferroelectric polarisation points towards one electrode or the other[15]. Second, the electrode's ground state can be modified by the interfacial accumulation of screening electrons[9,18,19]. This so-called ferroelectric field-effect[20] may be relevant for example if (at least) one the electrodes is a strongly correlated material, such as a manganite[9]. In addition, a debate exists on whether other

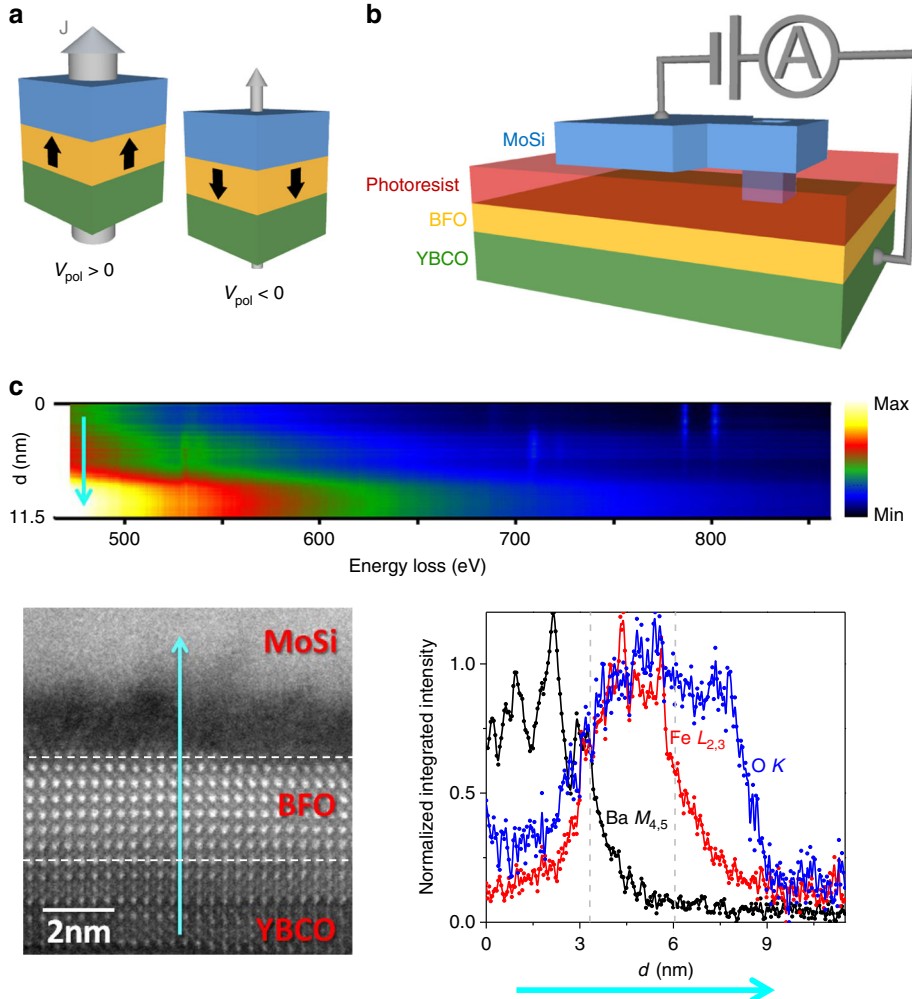

**Fig. 1 Junctions' structure. a** A tunnel junction whose barrier is ferroelectric can be reversibly switched between a high and low-conductance states by the application a voltage pulse of amplitude $V_{pol}$, which sets the remnant ferroelectric polarisation direction towards one electrode or the opposite. **b** Scheme of a superconductor/ferroelectric/superconductor tunnel junction based on a $YBa_2Cu_3O_{7-\delta}/BiFeO_3$ heterostructures on which a micrometric $Mo_{80}Si_{20}$ contact is made through an aperture across an insulating photo-resist overlayer. **c** (Top) EELS linescan acquired while scanning the electron beam across the stacking, moving a distance $d$ from the YBCO into the MoSi layer along the direction of the cyan arrow. The signal intensity is indicated by the colour scale (in arbitrary units). The edges of interest include the O $K$ edge near 528 eV, Fe $L_{2,3}$ with onset near 709 eV or the Ba $M_{4,5}$, near 781 eV. Principal Component Analysis was used to remove random noise. (Bottom, left) Atomic resolution HAADF image of the YBCO/ BFO/ MoSi stack. Then horizontal dashed lines highlight the approximate locations of the BFO top and bottom interfaces (Bottom, right) Normalised integrated intensities for the O, Fe and Ba signals in blue, red and black, respectively. The vertical dashed lines highlight the approximate locations of the BFO top and bottom interfaces. In all panels a cyan arrow shows the direction of the linescan.

mechanisms unrelated to ferroelectricity may contribute to the TER, for example the creation of conducting defects within the ferroelectric barrier due to ion electromigration[21,22].

The initial motivation of this work was studying the interplay between superconductivity and TER, using tunnel junctions composed of two superconducting electrodes that sandwich a ferroelectric, and across which superconducting Cooper pairs and quasiparticles may contribute to the conduction. As we will discuss later, the junctions were designed to boost TER mechanisms related to ferroelectric switching that we described above[9,15,18,19]. Unexpectedly, we found that none of these mechanisms are dominant.

In the following, we demonstrate that effects qualitatively and quantitatively identical to the TER are produced instead by a reversible electrochemical (redox) reaction that leads to oxygen exchange between the junction electrodes. This mechanism produces a giant TER in junctions that have a ferroelectric barrier, and also in junctions that consist of two dissimilar electrodes placed in direct contact—i.e. with no third material, ferroelectric or other, placed in between them. Because the redox reaction affects the physical properties of the electrodes, this scenario is different from that of junctions (or capacitors) in which the resistance switching is not in the tunnelling regime and is dominated by electromigration-induced changes within the thick insulating material placed between the electrodes[23–27]. The key tool to identifying electrochemistry as the physical mechanism at play is the study of ferroelectric and non-ferroelectric tunnel junctions in which one of the electrodes is a superconducting cuprate. This allows us to correlate the tunnel resistance switching with a modulation of the superconducting energy-gap at the junction interface. The use of a superconducting electrode brings up another key finding; the opening the superconducting energy-gap drastically increases the size of TER effects, which are up to 3000% larger when the current is carried by superconducting quasiparticles than when it is carried by normal electrons.

## Results

**Junctions' layout and structural characterisation.** The main experiments were carried out in $Mo_{80}Si_{20}$ (superconductor)/ $BiFeO_3$ (ferroelectric) /$YBa_2Cu_3O_{7-\delta}$ (superconductor) micron-size junctions [see scheme in Fig. 1b] fabricated through a combination of pulsed laser deposition, optical lithography and sputtering. $BiFeO_3$ (BFO) doped with Mn (5%) was chosen as ferroelectric barrier. The bottom electrode is $YBa_2Cu_3O_{7-\delta}$ (YBCO), an archetypal high-temperature superconductor with critical temperature $T_C \sim 90$ K (see supplementary information) on which BFO can be epitaxially grown[28]. This allows for high-quality heterostructures in which the modulation of super-conductivity by ferroelectric field-effects[20] has been demonstrated earlier[28]—a priori an interesting ingredient to promote TER. The top electrode, the amorphous alloy $Mo_{80}Si_{20}$ (MoSi), is a low-temperature superconductor ($T_C \sim 7$ K, see Supplementary Fig. 1). As we will see, the superconducting properties of MoSi do not play a role in the present experiments. YBCO and MoSi have very different carrier concentration—respectively, $\sim 3 \times 10^{20}$ cm$^{-3}$ and $\sim 3 \times 10^{22}$ cm$^{-3}$—[29,30] and consequently the Thomas-Fermi screening length is—expectedly[29] shorter in MoSi ($\sim$Å) than in YBCO ($\sim$nm). Thus, the pair of electrodes endow the junctions with the asymmetry that expectedly contributes to TER[15] of ferroelectric tunnel junctions. We studied junctions with fixed thickness of MoSi (100 nm) and YBCO (30 nm), and BFO thickness varying from 15 nm to 0 nm (that is, with MoSi and YBCO electrodes in direct contact). Additional control experiments (see Supplementary Figs. 3–5) were carried in junctions based on an insulating (non-ferroelectric) $SrTiO_3$ (STO)

interlayer instead of BFO, as well as with different counter-electrodes (Au and $In_2O_3/SnO_2$) instead of MoSi.

A typical scanning transmission electron microscopy (STEM) cross-section image of the junction's interfaces can be seen in Fig. 1c. Atomic resolution high angle annular dark field (HAADF) images show that samples grow epitaxially and the BFO/YBCO interface is coherent. The top BFO interface shows steps one-unit-cell high, resulting in some physical roughness which does not compromise the BFO layer integrity. The chemical composition can be measured from electron energy-loss spectroscopy (EELS). EELS linescans such as the one in Fig. 1c (top panel) can be obtained by scanning the electron beam across the interface while acquiring EEL spectra. The normalised integrated intensities corresponding to edges of interest such as O, Fe and Ba are shown in the bottom right corner. The O $K$ edge signal (in blue) is clearly detected on top of the BFO layer, whose location is depicted by the Fe $L_{2,3}$ edge (red). This O rich layer is extended 2–3 nm into the MoSi layer, which means that a highly oxidised, nanometric MoSi layer right is formed on top of the BFO surface. This finding is not surprising since cuprates have strong tendency to lose oxygen (due to the high reduction potential of $Cu^{+3}$) and, contrarily, MoSi has a strong tendency to oxidise (stronger than BFO) due to the negative reduction potential of Mo and Si (see Supplementary Table 1). Thus, one expects MoSi to spontaneously oxidise during its growth at the expense of reducing YBCO.

**Conductance experiments.** Figure 2a through 2c show the differential conductance $G = dI/dV_{BIAS}$ versus applied $V_{BIAS}$ at $T = 3.2$ K, for junctions with different BFO barrier thickness (see labels). The curves labelled ON were measured after application of a large "poling" voltage $V_{pol} > 0$ V, while the curves labelled OFF were measured after $V_{pol} < 0$ V. One can see that, for all samples, the conductance is very different in the ON and OFF states. In the ON state, the background dependence is nearly linear, and a "dip" is observed around zero bias. In the OFF state the background is roughly parabolic, and the conductance is down to two orders-of-magnitude lower than in the ON state. Indeed, for the sample with thicker BFO (Fig. 2c) the conductance in the OFF state is below the experimental resolution ($\sim 10^{-8}$ S) within the explored bias range.

Figure 2d–f display the conductance under bias $V_{BIAS} = 100$ mV and $V_{BIAS} = 0$, namely $G_{100}$ and $G_0$, measured in the remnant state after application of different $V_{pol}$. This is cycled from negative to positive and vice-versa in order to repeatedly switch between the high (ON) and low (OFF) conductance states. The switching is hysteretic, bipolar, and reversible. $G(V_{pol})$ loops are reproducible: each of them contains data from subsequent cycles, which appear superposed. Repeated switching (up to at least one hundred times) produces no significant variation of the ON/OFF conductance levels. Notice that full switching between the two states is obtained for $|V_{pol}|$ in the few-volts range, and that the loops are asymmetric. That is, a lower $|V_{pol}| \sim 1$ V is required to switch from the ON into the OFF state than vice versa, for which $V_{pol} \sim 3$ V. As shown in Supplementary Fig. 4, similar results (size of the resistance switching and switching voltage values) are obtained when the BFO interlayer is substituted by $SrTiO_3$ (a non-ferroelectric band insulator) of similar thickness. However, when MoSi is substituted by a noble-metal (Au) in direct contact with YBCO, the resistance switching is orders-of-magnitude weaker (see Supplementary Fig. 3).

Note in Fig. 2d, e that the largest resistance switching is observed for low $V_{BIAS}$. For example, in Fig. 2d, one finds that the electroresistance (defined as $ER \equiv G^{ON}/G^{OFF}$) is $ER_0 \sim 5000$ at zero bias (see the $G_0$ loop, dark circles), while it is $ER_{100} \sim 150$ at

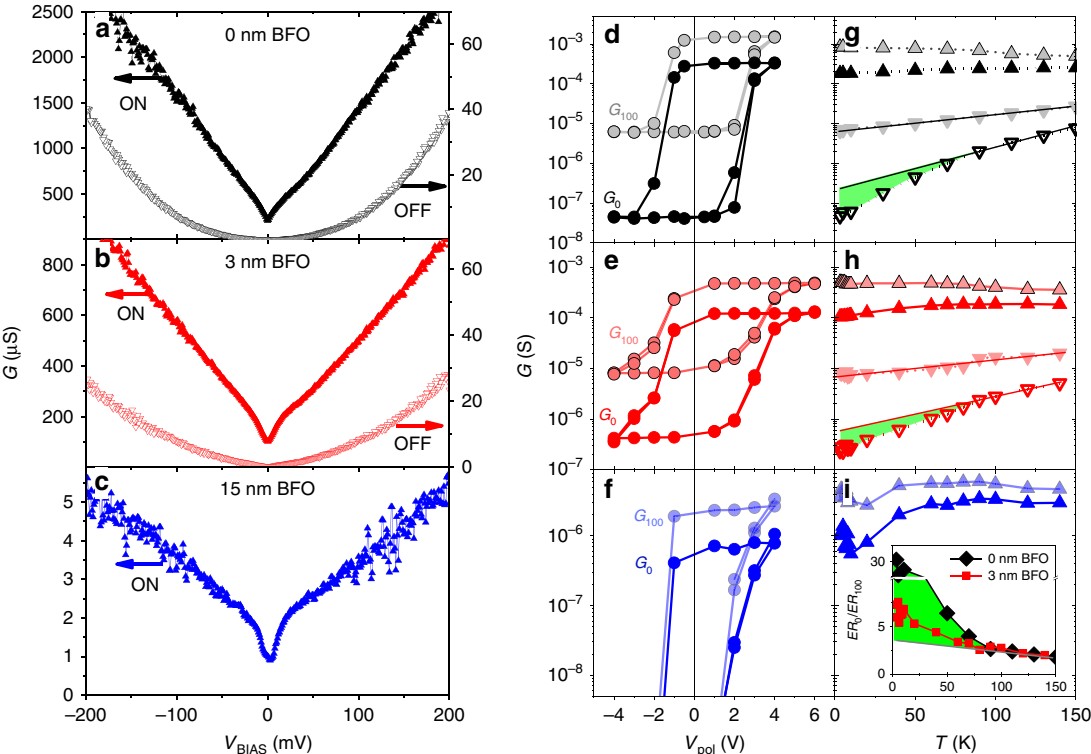

**Fig. 2 Conductance switching.** Differential conductance $G = dI/dV_{BIAS}$ as a function of applied $V_{BIAS}$, measured at $T = 3.2$ K after application of $V_{pol} > 0$ V (ON state) and $V_{pol} < 0$ V (OFF state) for three different BFO thickness. **a** 0 nm BFO-Mn ($V_{pol} = \pm 3$ V), **c** 3nmBFO-Mn ($V_{pol} = +6$ V and $-4$V) and **e** 15nmBFO-Mn ($V_{pol} = \pm 4$ V). For the latter thickness, the conductance in the OFF state is unmeasurably low. **d**, **e**, **f** show the switching between the ON and OFF state. This is illustrated by $G_0(V_{pol})$ and $G_{100}(V_{pol})$, that is, the conductance measured at zero bias and at $V_{BIAS} = 100$ mV after application of different $V_{pol}$, which were cycled repeatedly from negative to positive and vice versa. For the same samples as in **a**–**c**. **g**–**i** show $G_0(T)$ and $G_{100}(T)$ after poling at 3.2 K. The straight lines superposed to the measurements in the OFF state are a extrapolation of the high temperature trend of $G(T)$. Green regions highlight the deviation of $G_0(T)$ from the high temperature trend. The inset in (i) shows the ratio between the electroresistance ($ER = G_{ON}/G_{OFF}$) at zero bias and that under $V_{BIAS} = 100$ mV, that is $ER_0/ER_{100}$, as a function of temperature for the 0 nm BFO (diamonds), and 3 nm BFO junction (squares symbols). $ER_0/ER_{100}$ is calculated from the data in 2 g and 2 h. The green region highlights the deviation of $ER_0/ER_{100}$ from the high temperature trend. The vertical axis presents a break between $ER_0/ER_{100} = 10$ and $ER_0/ER_{100} = 25$.

$V_{BIAS} = 100$ mV (see the $G_{100}$ loop, light-coloured circles). The same bias dependence is evident in Fig. 2e.

Experiments as those described above were conducted for series of junctions with varying dimensions. Fig. 3 shows the relationship between $G_{100}$ (in the ON and OFF states) and the junctions' area $A$. In the OFF state, the conductance is directly proportional to $A$ (note that the hollow triangles in Fig. 3a, b display constant $G_{100}/A$ values), while in the ON state it is not, as already observed in ferroelectric tunnel junctions[31,32]. In fact, in the ON state the conductance is directly proportional to the junctions' perimeter $P$ (note that solid triangles in Fig. 3d–f display constant $G_{100}/P$ values). The scaling observed in the OFF state suggests homogeneous conduction over the junction area, which is consistent with electron tunnelling and rules out current-shunting through conducting defects, such as filaments or pinholes[6]. The scaling observed in the ON state implies that the resistance switching occurs only over the junction's periphery. As shown earlier[21,31], this can be understood by considering that the electric field produced upon application of $V_{pol}$ is stronger over the edges than in the central area of the junction, favouring the local activation of the switching mechanism. Note that this explanation applies to any electric-field activated switching mechanism[21,31]. For instance, if the mechanism is based on the switching of the ferroelectric polarisation, the argument is that the polarisation reverses only over the junction' periphery, but remains pinned everywhere else[31]. Finally, from the quantitative point of view, it is worth noticing that the conductance levels

$G_{100}/A \sim 10^{-8}$ S$\mu$m$^{-2}$ (OFF state) and $G_{100}/P \sim 10^{-5}$ S$\mu$m$^{-1}$ (ON state) of the 0 nm and 3 nm BFO junctions are comparable to those observed in standard ferroelectric tunnel junctions[31].

In summary, the observed resistance switching characteristics are globally as expected from the extensive literature on TER in ferroelectric tunnel junctions, both qualitatively (shape and symmetry of the switching loops, reversibility, endurance) and quantitatively (switching voltages, size of the conductance switching, scaling with the junctions' size). However, because the behaviour is similar in junctions in which YBCO and MoSi are in direct contact (Fig. 2a, d), in junctions with a BFO interlayer, and in junctions with a non-ferroelectric SrTiO$_3$ interlayer (see Supplementary Fig. 4), it is evident that ferroelectricity does not play a central role in it. From the phenomenological perspective, there is only one key dissemblance between the observed effects and the TER of oxide-based ferroelectric tunnel junctions: here the electroresistance shows a strong bias dependence which, as revealed by the temperature behaviour discussed below, is connected to superconductivity.

Fig. 2g–i show $G_{100}$ (light symbols) and $G_0$ (dark symbols) as a function of temperature, both in the ON (up triangles) and OFF state (down triangles, not measurable for the thickest BFO sample). Notice that the ON and OFF states are set by applying the required $V_{pol}$ at 3.2 K and the remnant conductance is subsequently measured for increasing temperatures. The temperature dependence of the conductance presents significant differences between different measuring bias. For high bias

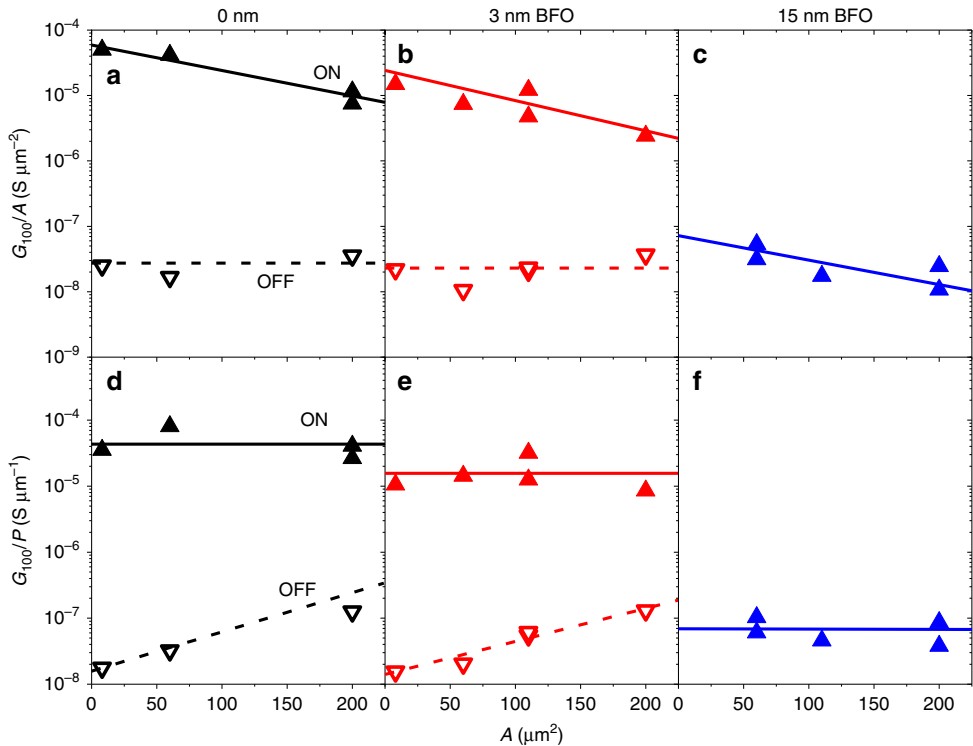

**Fig. 3 Scaling of the conductance with the junctions' size.** For a series of junctions with variable area $A$: **a**, **b**, **c** $G_{100}$(3.2 K) in the ON and OFF states normalised by the junction area, $G_{100}/A$; and **d**, **e**, **f** $G_{100}$(3.2 K) in the different states normalised by the junction perimeter, $G_{100}/P$ for the different samples. Lines are a guide to the eye.

($G_{100}$, light symbols), the behaviour is similar to that of ferroelectric tunnel junctions with normal-metal electrodes[32]: in the ON state, the conductance is nearly constant or slightly decreases with increasing temperature, while in the OFF state it exponentially increases with increasing temperature (see the straight line superposed to the data in Fig. 2g, h). The zero-bias conductance ($G_0$, dark symbols) and $G_{100}$ behave similarly at high temperatures. However, below a given temperature, $G_0$ drops below the level expected from the high temperature trend. This departure is weaker in the ON state (see Supplementary Fig. 2 for further details) than in the OFF state, for which a pronounced deviation from the exponential dependence (straight line) is observed below ~70 K (departure highlighted by the green regions). The above observations imply that, the lower the temperature, the larger the electroresistance $ER \equiv G^{ON}/G^{OFF}$, the enhancement being stronger when measured at zero bias than when measured at 100 mV. Indeed, at low temperatures $ER_0 \gg ER_{100}$. This is clearly observed in the inset of Fig. 2i, which displays $ER_0/ER_{100}$ as a function of temperature for junctions with a 0 nm (black) and 3 nm (red) BFO interlayer. Both curves show a clear upturn below $T \sim 70$ K, which is more pronounced for the junction with no BFO interlayer (black symbols). For this junction, at the lowest temperature, the zero-bias electroresistance $ER_0$ is 30 times larger than $ER_{100}$. As demonstrated below through Figs. 4 and 5, this behaviour is explained by the opening of the superconducting energy-gap in YBCO.

Fig. 4a, d show the normalised conductance $g(V_{BIAS}) \equiv G(V_{BIAS})/G(60$ mV) of the 3 nm BFO sample for different temperatures, respectively, in the ON and OFF states. The overall behaviour corresponds to the one typically observed in contacts between a metallic electrode and c-axis YBCO[33,34], and is as expected for quasiparticle tunnelling across a low-transparency interface (see Supplementary Note 2 for further details). At low temperatures, a "dip" is observed in the low-bias range, in which

the conductance drops below the background due to the opening of the superconducting energy-gap[35]. This gap feature is clearer in Fig. 4b, e, which, respectively, display the derivative $dg/dV_{BIAS}$ of the data in 4a and 4d. In both sets of curves (Fig. 4b, e) the gap feature clearly stands out from the nearly linear background observed at high temperatures and bias. To better determine the temperature and bias range in which that gap-feature develops, Fig. 4c, f show a colour plot of the same data as in 4b and 4e, represented in a different fashion: we plot the ratio between the derivative at the highest $T = 140$ K, $(dg/dV_{BIAS})^{140 K}$, and the derivative at each temperature $T$, $(dg/dV_{BIAS})^T$. Considering the colour scale, green means no deviation from the conductance trend observed at high temperature. The gap-feature stands out in brown colour indicating the conductivity suppression due to the opening of the gap. Direct comparison between Fig. 4c (ON state) and 4f (OFF state) evidences that the superconducting gap opens at lower temperature and within a narrower bias range in the OFF than in the ON state. This data representation allows us to estimate the critical temperature $T_C$ (horizontal dashed line) and gap size $\Delta$ (vertical dashed lines). In the ON state, we find $T_C \sim$ 90 K and $\Delta_{YBCO} \sim 25$ meV, which correspond to the values expected for optimally doped YBCO. However in the OFF state, $T_C \sim 70$ K and $\Delta_{YBCO} \sim 20$ meV, which correspond to the values for underdoped YBCO. In conclusion, the conductance in the ON state corresponds to tunnelling into optimally doped YBCO, while in the OFF state we observe tunnelling into YBCO with depressed superconducting properties. Thus, the switching between the high (ON) and low (OFF) tunnelling conductance states, driven by the application of $V_{pol}$, is accompanied by a modulation of the superconductivity in YBCO. Note that the present measurements could not resolve spectral features related with superconductivity in the MoSi electrode, whose energy-gap ($\Delta_{MoSi} \sim 1$ meV) is one order-of-magnitude narrower than YBCO's.

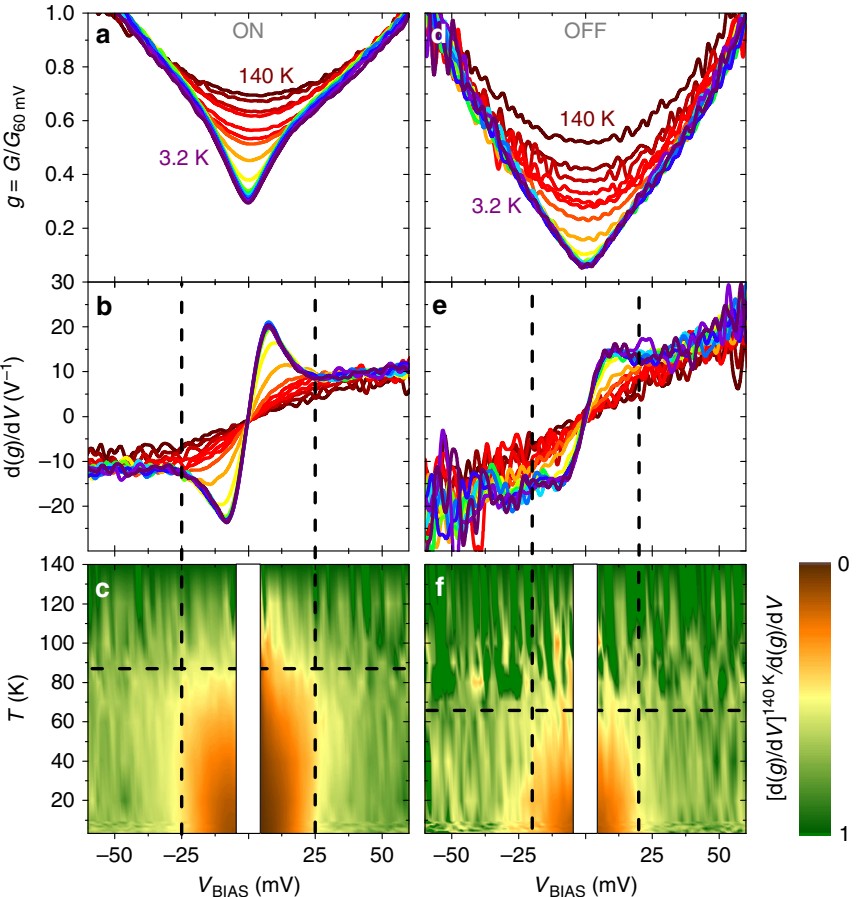

**Fig. 4 Temperature dependence of the spectral features.** For a 3 nm BFO junction, in the ON state (**a**) and OFF state (**d**): differential conductance normalised to the conductance at 60 mV, $G(V_{BIAS})/G_{60\,mV}$, for different temperatures in the series 3.2, 4, 5, 6, 7, 8, 9, 10, 20, 40, 60, 80, 90, 100, 120, 140 K, which correspond to the different curve colours. The series extrema are indicated by the labels. **b**, **e** respectively, show the first derivative $dG/dV_{BIAS}$ of those curves. **c**, **f** shows the ratio between the $dg/dV_{BIAS}$ at $T = 140$ K and $dg/dV_{BIAS}$ at any other temperature as a function of the bias voltage and temperature. The range $V_{BIAS} < 5$ mV is masked because the ratio between two values close to zero yields a large data scattering. The vertical dashed lines show the bias span of the gap feature, the horizontal dashed lines point the critical temperature at which the gap starts opening.

**Modelling the electroresistance increase below $T_C$.** The opening of superconducting gap in YBCO leads to a much larger electroresistance $ER \equiv G^{ON}/G^{OFF}$ when measured around zero bias than when measured at 100 mV (compare the loops for $G_0$ and $G_{100}$ in Fig. 2d, e), and to the increase of the ratio $ER_0/ER_{100}$ below $T \sim 70$ K (inset of Fig. 2i). This is because, near zero bias, a significant part of the tunnelling current is carried by quasiparticle excitations. These gradually disappear as the temperature $T$ is decreased below $T_C$, making the zero-bias conductance $G_0$ drop. This is illustrated by generalised Blonder-Tinkham-Klapwijk (BTK) theory[8,9] simulations of the junction conductance vs. bias at different temperatures, which are shown in Fig. 5a—see the calculation details in Supplementary Note 3. Notice that both $G_0^{ON}$ and $G_0^{OFF}$ drop with decreasing temperature, as it can be seen in the BTK simulations shown in Fig. 5b, but the drop is more pronounced in the OFF state because the junction's transparency is lower than in the ON state[36]. As a consequence, $ER_0 \equiv G_0^{ON}/G_0^{OFF}$ gradually increases as the temperature is decreased below the $T_C$ in the OFF state. At variance, for $eV_{BIAS} = 100$ meV $>> \Delta_{YBCO}$ the tunnelling current is essentially carried by electrons and therefore $G_{100}$ is unaffected by the quasiparticles population. This results in a weak temperature dependence for $ER_{100}$, as observed[32] in standard ferroelectric tunnel junctions. All of the above explains the increase of $ER_0/ER_{100}$ observed in the experiments below $T \sim 70$ K (inset in

Fig. 2i), which is well captured by the BTK simulations shown in the inset of Fig. 5b.

**Origin of the resistance switching.** A model of the resistance switching mechanism that accounts for the overall experimental findings is sketched in Fig. 6, which displays a cartoon of the junction's interfaces. The model considers oxygen exchange between YBCO and the junction materials through a reversible redox reaction triggered by $V_{pol}$. The asymmetry in the switching $V_{pol}$ (Fig. 2d–e and Supplementary Fig. 5) implies that different energy barriers are overcome depending on whether the junction is driven from OFF into ON or vice versa.

Let us start with the case in which there is direct contact between YBCO and MoSi. In the low-conductance state (OFF, Fig. 6a) MoSi is oxidised at the interface (this is labelled MoSiO$_x$) at the expense of leaving behind oxygen vacancies in the YBCO (the oxygen-deficient YBCO is labelled YBCO$_{1-x}$). This is the system's ground state because, as discussed above, YBCO is the electrode with the highest reduction potential (+2.4 V, see Supplementary Table 1). In this state, the interfacial YBCO is severely deoxygenated, which reduces the interface transparency for electrons and quasiparticles, yielding a low junction conductance. The oxygen-deficiency extends beyond the interface, yielding a layer with depressed superconducting properties,

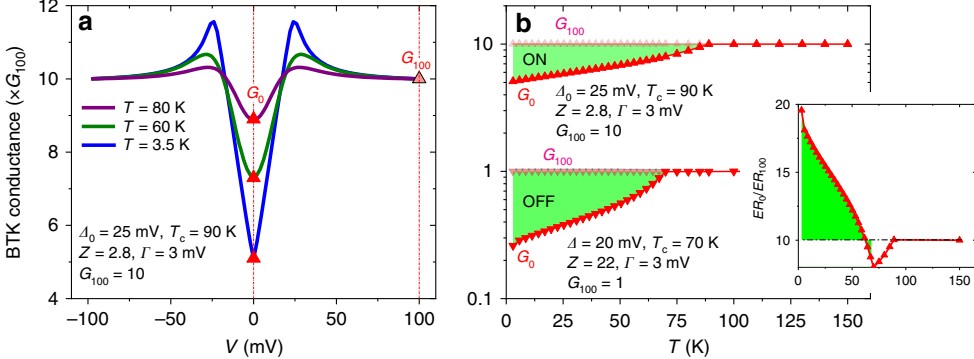

**Fig. 5 BTK modelling of the temperature dependent quasiparticle tunnel electro-resistance. a** Example of the BTK conductance *vs.* bias at different temperatures for a set of parameters (indicated in the legend) that correspond to the ON state. $G_0$ and $G_{100}$ are respectively indicated by the dark/light triangles. **b** $G_0(T)$ (dark triangles) and $G_{100}(T)$ (light triangles) calculated in the ON and OFF state using the BTK parameters indicated in the legend. Notice that $G_0(T)$ gradually drops below $G_{100}(T)$ as temperature is decreased beyond $T_C$, which is highlighted by the green regions. The departure is more pronounced in the OFF state than in the ON state. The inset shows the ratio $ER_0/ER_{100}$ as deduced from the calculations shown in the main panel. Notice the steady increase below the critical temperature in the OFF state ($T_C = 70$ K), highlighted by the green region.

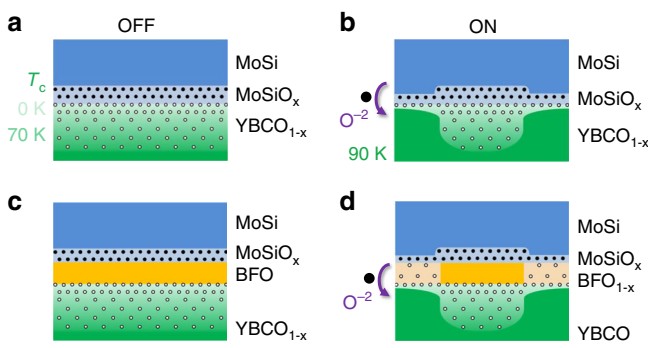

**Fig. 6 Electro-resistance model based on electrochemistry.** Schematic representation of the junctions with and without BFO. The oxygen anion ($O^{-2}$; black dots) and oxygen vacancy ($V_O{}^{+2}$; hollow dots) distributions are indicated. **a** OFF state of the 0 nm BFO junctions. The bottom electrode shows a gradual oxygen depletion (and thus depressed superconducting properties) which is very strong near the interface due to the interfacial oxidation (MoSiO$_x$) of the top MoSi electrode. **b** By applying $V_{pol} > 0$, $O^{-2}$ migrate downwards and the oxygen depletion is much more confined near the interface, leaving optimally doped YBCO. This occurs on the junction's edges, where the electric field is stronger. **c** OFF state of the junctions with BFO, in which YBCO shows gradual oxygen depletion (and thus depressed superconducting properties) which is very strong near the interface. In this state BFO is not significantly oxygen deficient. **d** By applying $V_{pol} > 0$ (ON state), oxygen vacancies $V_O{}^{+2}$ migrate into de BFO, resulting in a fully oxygenated YBCO and in oxygen-depleted, conducting BFO.

as revealed by the smaller energy-gap and low $T_C$ deduced from the tunnelling spectra (Fig. 4f). Upon application of a sufficiently high $V_{pol} > 0$ (Fig. 6b), the redox reaction as reversed. For this process, the highest energy barrier to be overcome—which determines $V_{pol}$ in a first approximation—is given by the difference between the reduction potentials $E_0$ of MoSi and YBCO. This difference is $\Delta E_0 \sim 2.6$–$3.2$ V (see Supplementary Table 1), largely dominated by the high reduction potential of YBCO. This is in good agreement with the switching $V_{pol}$ observed experimentally. The induced redox reaction significantly thins down the oxidised MoSiO$_x$ and oxygen-deficient YBCO$_{1-x}$ layers, with two consequences. First, it yields higher interface transparency and thus higher tunnelling conductance. Second, the YBCO layer becomes optimally doped close to the interface, as shown by $\Delta \sim 25$ meV and $T_C \sim 90$ K deduced from the tunnelling spectra (Fig. 4c). Application of a $V_{pol} < 0$ returns the

junction to back to its ground state (OFF). Notice that in this case the redox reaction is spontaneous from the electrochemical point of view. Thus the only energy barrier to be overcome by $V_{pol}$ is the barrier for ion transport. The fact that all of the junctions show the same ON to OFF switching $V_{pol} \sim -1$ V, regardless of the presence of an interlayer and of the top electrode material (see Supplementary Fig. 5), suggests that the barrier for ion transport arises at the interfacial YBCO layer.

The same electrochemical mechanism can explain the similar effects observed in the presence of a BFO interlayer, only in this case the oxygen exchange that results in the OFF/ON switching involves ion transport across this material. As discussed above and shown by the microscopy (Fig. 1c), an oxidised MoSiO$_x$ layer naturally form at the interface, which occurs at the expense of leaving oxygen vacancies in the YBCO because this is by far the material with the highest reduction potential in the stack (see Supplementary Table 1). The switching into the ON state (Fig. 6d) is produced upon application of $V_{pol} > 0$, which moves oxygen back into YBCO. The required $V_{pol} \sim 3$ V is similar as in the absence of BFO interlayer, because it is dominated by the high reduction potential of YBCO. Note however that the reduction potential of BFO is higher than that of MoSi (see Supplementary Table 1). Thus, besides thinning the MoSiO$_x$, the OFF to ON switching process likely involves the creation of oxygen vacancies in BFO, which therefore becomes conducting[24] and contributes to enhancing the junction conductance. Application of $V_{pol} \sim -1$ V drives the junction back to its ground state (OFF) as the barrier for ion transport in YBCO is overcome, allowing for the reaction that oxidises BFO and MoSi, and leads to YBCO with depressed superconducting properties near the junction interface. In this scenario, as expected, the conductance becomes unmeasurably low in the OFF state for the thicker BFO junctions (Fig. 2c–i). Notice that, in all cases, oxygen exchange occurs only over the periphery of the junctions, where the electric-field (that assists the redox reaction by activating ion motion) is stronger[21,31]. This explains the perimeter scaling observed in Fig. 3. Further quantitative analysis is shown in Supplementary Figs. 6 and 7, including fits of the normal-state tunnel conductance for estimates of the effective tunnel barrier thickness $d$ and height $\phi$. Those yield $d \sim 5$ nm and $\phi \sim 0.15$ eV for junctions with 0 nm BFO and, consistently, $d \sim 8$ nm and $\phi \sim 0.53$ eV for junctions with 3 nm BFO. In absence of BFO, the $\phi \sim 0.15$ eV barrier is due to the decrease in YBCO work function resulting from the electron doping by the oxygen vacancies. The average barrier height $\phi \sim 0.53$ eV in samples with BFO is explained by

the difference of the work functions of electron-doped BFO (~4.7 eV)[37] and YBCO (~5.2 eV)[38].

## Discussion

While redox reactions and the resulting changes in the electrodes' oxidation state may not be the dominant TER mechanism in many of the ferroelectric tunnel junctions studied in the literature, the present study shows that electrochemistry can account for the TER in some cases, in particular if the reduction potential of the involved materials is very different. For instance, materials, such as manganites and other transition metal oxides, may be prone to reduction when combined with low reduction potential metallic electrodes (e.g. Al or Co). In any case, we show here that functional characteristics of the TER (e.g. the magnitude of the effects, reversibility, endurance, scaling with the device size) and particularly those that make it unique as compared to other resistive switching phenomena (conduction in the tunnelling regime and concomitant manipulation of the electrode's ground state), can be obtained without the use of ferroelectrics if the junction's electrodes are judiciously chosen. Furthermore, in this case the strongest effects can be achieved if the electrodes are in direct contact, without any barrier material (ferroelectric or other) placed between them.

The fact that this demonstration is based on superconductors is interesting, beyond the fact that it allows in operando clear-cut spectroscopic evidence of the oxidation state, because it opens the door to realising Josephson effects (tunnelling of Cooper pairs) that can be switched between non-volatile states by voltage pulses. These would be of much interest in the field of superconducting electronics[39]. While in the present experiments the weak Josephson offered by YBCO—due to its short coherence length along the c-axis— and the high junction resistances have precluded the observation of such effects, extensions of this work in which the junction interface is along other YBCO crystallographic directions should provide access to them.

## Methods

**Sample fabrication**. Heterostructures with fixed YBCO thickness (30 nm) and variable BFO (0 to 15 nm) were grown on SrTiO$_3$ (STO) substrates by pulsed laser deposition, using a KrF 248 nm excimer laser with an energy density of ~1 Jcm$^{-2}$, and a repetition rate of ~5 Hz. The BFO target was doped with 5% Mn, which has been shown to reduce leakage currents in ultrathin films[40]. The homogeneity of the heterostructures was ensured by using a rotating substrate holder. The growth temperatures were 700 °C for YBCO and 560 °C for BFO. A pure oxygen atmosphere (0.36 mbar) was maintained during the subsequent deposition (in situ) of both materials. After BFO growth, the samples were cooled-down in a pure oxygen atmosphere (800 mbar), searching of an optimum oxygen stoichiometry. STEM measurements were carried out in an aberration-corrected JEOL ARM200cF electron microscope operated at 200 kV equipped with a cold field emission gun. Samples were prepared by conventional mechanical grinding and Ar ion milling. Vertical junctions were defined by standard photolithography. A first photoresist layer is spin-coated on top of the heterostructures, in which an array of square holes (that define the junction area $A$, in the range 10–200 μm$^2$) is patterned. Following this, the photoresist is hard-baked, so that it becomes immune to subsequent illumination, developing and lift-off processes, and thus permanent. A second photoresist is then spin-coated in order to define an electrical contact pad on top of each of the square holes. An oxygen plasma was used to eliminate all resist residues from the YBCO (or BFO) surface prior to sputtering deposition of a 100 nm thin MoSi film at room temperature, which is followed by the resist lift-off.

**Transport measurements**. For transport measurements, the voltage applied to the junction is first ramped to $V_{pol}$ (typically up to a few volts) and back to zero. Subsequently, $I(V_{BIAS})$ characteristics are measured at lower voltages ($|V_{BIAS}| < 200$ mV). A numerical derivative was performed in order to obtain the conductance spectra. In all cases, the top electrode (MoSi) was grounded and the voltage was applied to the bottom electrode (YBCO).

## Data availability

The data that support the plots within this paper and other findings of this study are available from the corresponding author upon reasonable request.

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

## Acknowledgements

Work supported by the ERC grant N° 647100 "SUSPINTRONICS", French ANR grants ANR-15-CE24-0008-01 "SUPERTRONICS", and ANR-16-CE24-0028-01 "QUANTU-MET", and European COST action 16218 "Nanocohybri". Work at UCM supported by Spanish MINECO-FEDER MAT2015-66888-C3-3-R and ERC PoC2016 POLAR-EM and Quantox of QuantERA ERA-NET Cofund in Quantum Technologies (Grant Agreement N. 731473). V. R. acknowledges the European Union's Horizon 2020 research and innovation programme (Marie Skłodowska-Curie IF grant agreement OXWALD n° 838693). J.S. thanks INP-CNRS and "Scholarship program Alembert" funded by the IDEX Paris-Saclay ANR-11-IDEX-0003-02 for support during his stay at the Unité Mixte de Physique CNRS/Thales. We thank V. Garcia, S. Fusil and A. Barthélémy for discussions and critical reading of the paper.

## Author contributions

The study was designed and supervised by J.E.V. The oxide/metal heterostructures were grown by A.S., J.B. and S.C. J.G. and M.V. carried out the electron microscopy studies. The junctions were fabricated by V.R. The electrical measurements were carried and analysed by V.R. with the help of R.E.H. The BTK simulation were performed by K.S. and X.P., with the support of A.I.B, J.L. and J.E.V. The electrochemical model was proposed by V.R. and developed with C.L. and J.S. The results were discussed and interpreted by all the authors cited so far, with contributions from J.T., K.B., G.S., N.B. and C.F-P. The paper was written by J.E.V. and V.R. with contributions from all the authors.

## Competing interests

The authors declare no competing interests.
