## [Peer Review File · Nature Communications]

Editorial Note: This manuscript has been previously reviewed at another journal that is not operating a transparent peer review scheme. This document only contains reviewer comments and rebuttal letters for versions considered at Nature Communications .

Reviewers' Comments:

Reviewer #2:

Remarks to the Author:

The authors did a great job in addressing the issues raised by all the reviewers, which were remarkably similar. The claims of the paper are indeed much better justified. However I am still not impressed by the quantitative analysis. Indeed several numbers have been checked which is good, (why only in supplementary?), but for a quantitative analysis, I would expect a theoretical understanding of the main findings, i.e. of the factor 30 increased on/off ratio for the superconducting state. Why don't the authors attempt to theoretically reproduce the main features in Figs. 2d-i?

I also have a few minor comments as listed below.

I still doubt the relevance of the paper for a very broad audience, as the authors try to argue in their response to the reviewer comments. However, since the new submission is to Nature Communications, which scope is to publish major findings in a specific field, I think this is the appropriate journal for this manuscript. I would therefore recommend publication in NComms if the authors can take away the main concern above.

In the abstract, the sub-sentence 'those results ... tunneling regime' sounds empty to me. If it cannot be made more concrete the text would benefit from removing it.

P3 halfway typo: carrier -> carried

P3 3rd paragraph: carried -> carried out

P4 sentence about screening length: the statement doesn't make sense now, in brackets the MoSi length is shorter.

Reviewer #3:

Remarks to the Author:

The authors have made significant changes to the manuscript, including more data and control experiments, as well as theoretical fits using the BDR and BTK models (normal state and superconducting states, respectively). This has certainly strengthened the work, and I would tentatively state that it might be suitable for publication in Nature Communications. However there are still a number of issues that should be addressed in my opinion, mostly associated with the data analysis and explanations.

Fundamentally I am still perplexed by the authors flow of the argument and focus on the samples with the ferroelectric (FE) layer of BFO: they make a clear story of the difference between their data and others since they observe this effect in the absence of a FE layer, but then go on to focus on the BFO samples. This can probably be easily fixed by rewording the introduction.

Regarding the new data and theoretical fits: the STEM EELS seems to show an *enhanced* oxygen concentration near the BFO on the YBCO side. Isn't this in contradiction to statement that that YBCO will naturally oxygen deplete at the interface during MoSi growth (top of page 5)?

I am still confused about the similar values for the switching fields in the very different cases (no BFO

and thick BFO) – surely the voltage drop across the heterostructure mesa is very different in the case when a highly resistive barrier is used, and one might expect the electrochemistry to occur at quite different voltages. Of course there is the caveat of the fringing fields around the perimeter being important, but still I think my argument holds. In this context I don't understand the comment in section 1 of the supplementary "The voltage required to reverse... is dominated by the large E_0 for YBCO", and a related statement in the main text at the top of page 10.

On page 11 they state that the barrier for ion transport arises at the YBCO interfacial layer. Is this obvious? Why is it not at the MoSi interface?

The BTK fits are poor, and I do not have confidence in the values extracted from these fits (for a gap or transparency). Similarly the discussion of the BDR fit was very brief (the equation fitted too, with any assumptions, was not shown, and the errors on the extracted parameters were not stated). It would be nice if these values could be referred to in the main text, and used to build a band alignment picture of these devices. These Supplementary data are measured at what temperature? There is a lack of information in many of the figure captions in this regard.

Minor details:

The statement that the Fermi screening length is expectedly *shorter* in the YBCO compared to the MoSi should be *longer*.

The authors should state their lower measurement limit of conductivity ("unmeasurably low", on page 5).

Figure 1: could the authors put some indication on the TEM cross-section of the location of the chemical interfaces (as they have done in the integrated intensity vs depth plot using grey dashed lines).

Reviewer #4:

None

Response to reviewer #2 (Remarks to the Author):

We thank the referee for reviewing our manuscript once again. We are glad that he/she finds we did a great job in addressing the issues raised by all the reviewers, and that the claims of the paper are now much better justified. Yet in her/his report, the referee asks for additional improvements. We have considered all of her/his new requests in full, as described below. We hope that the referee finds the new version of our paper suitable for publication in Nature Communications.

Referee's comment # 1: *“The authors did a great job in addressing the issues raised by all the reviewers, which were remarkably similar. The claims of the paper are indeed much better justified. However, I am still not impressed by the quantitative analysis. Indeed, several numbers have been checked which is good, (why only in supplementary?), but for a quantitative analysis, I would expect a theoretical understanding of the main findings, i.e. of the factor 30 increased on/off ratio for the superconducting state. Why don't the authors attempt to theoretically reproduce the main features in Figs. 2d-i?”*

Response: To comply with the referee's request, we have refined the theoretical analysis, which has allowed us to reproduce the main features in Figs. 2d-i.

In essence, we have performed calculations based on the argument already given in the previous version of the manuscript: that the large increase of the low-bias ER as temperature decreases below T_c is caused by the decrease of quasiparticle excitations, which leads to a decrease of the zero-bias conductance that is faster in the OFF than in the ON state.

In order to numerically simulate the above scenario, we have used the BTK theory for the conductance across S/N interfaces with finite barrier strength Z . Following the existing literature for d-wave superconductors, we have improved the BTK simulation by considering the finite quasiparticle lifetime, a spatial inhomogeneity in the energy-gap, as well as by removing the bias dependence of the background conductance (evident for $eV > \Delta$) that is not accounted for by the BTK model. These precautions have allowed us to obtain simulations much closer to the experimental data, and to extract the parameters Z in the ON and OFF states. From this, and always using the BTK theory, we have been able to theoretically reproduce the key experimental features, as requested by the referee, in particular the drastic increase of the electroresistance for $T > T_c$.

Changes made: The analysis outlined above and the key conclusions are now referred to in the main text (see page 9):

“The opening of superconducting gap in YBCO leads to a much larger electroresistance $ER \equiv G^{ON}/G^{OFF}$ when measured around zero bias than when measured at 100 mV (compare the loops for G_0 and G_{100} in Figs. 2d and 2e), and to the increase of the ratio ER_0/ER_{100} below $T \sim 70$ K (inset of Fig. 2i). Near zero bias a significant part of the tunnelling current is carried by quasiparticle excitations. These gradually disappear as the temperature T is decreased below T_c , making the zero-bias conductance G_0 drop (this is illustrated by Fig. SI.9). That applies both to G_0^{ON} and G_0^{OFF} , but the drop is more pronounced in the OFF state because³⁷ the junction's transparency is lower than in the ON state (see Figs. SI.8 and SI.9 for quantitative estimates). As a consequence, $ER_0 \equiv G_0^{ON}/G_0^{OFF}$ gradually increases as the temperature is decreased, particularly below the T_c in the OFF state. At variance, for $eV_{BIAS} = 100$

$meV \gg \Delta_{YBCO}$ the tunnelling current is essentially carried by electrons and is therefore unaffected by the quasiparticles population. This results in a weak temperature dependence for ER_{100} , as observed³³ in standard ferroelectric tunnel junctions. All of the above explains the increase of ER_0/ER_{100} observed below $T \sim 70$ K (inset in Fig. 2i), which is further illustrated by the simulations in Fig. SI. 9.”

In addition, we provide a detailed explanation of the theoretical simulations of the low temperature conductance curves as well as various new figures in the Supplemental Information (sections 7 and 8)

Referee’s comment # 2: “In the abstract, the sub-sentence ‘those results ... tunnelling regime’ sounds empty to me. If it cannot be made more concrete the text would benefit from removing it.”

Response: We have removed that sentence from the abstract.

Referee’s comment # 3:

“P3 halfway typo: carrier -> carried

P3 3rd paragraph: carried -> carried out

P4 sentence about screening length: the statement doesn’t make sense now, in brackets the MoSi length is shorter.”

Response: Thank you for pointing out these errors. They have been corrected.

Response to reviewer #3:

We thank the referee reviewing our manuscript once again, and for acknowledging that we have made significant changes that have strengthened our work. Yet in her/his new report, the referee asks for further revision associated with the data analysis. We have considered all of the suggestions and criticisms, and we have revised the paper accordingly as detailed below. We hope that, after considering our responses and the changes made, the referee finds the revised version suitable for publication.

***Referee's comment # 1:** "Fundamentally I am still perplexed by the authors flow of the argument and focus on the samples with the ferroelectric (FE) layer of BFO: they make a clear story of the difference between their data and others since they observe this effect in the absence of a FE layer, but then go on to focus on the BFO samples. This can probably be easily fixed by rewording the introduction".*

Response: The reason why we studied BFO samples is, as the referee may anticipate, that we were looking for ferroelectric effects. We think that discussing the initial motivation for the experiments is the most honest and informative for the reader. Indeed, BFO samples turn out to be very important: they demonstrate that electrochemical effects may be dominant even if, as explained in the introduction, giant electroresistance effects associated to ferroelectric switching were expected from the presence of the three key ingredients i) a ferroelectric barrier, ii) electrodes with dissimilar screening length and iii) ferroelectric field effects having been observed earlier at the YBCO/BFO interface.

Changes made: We have tried to make things clearer by rewording the introduction, as suggested by the referee. Now the paragraph reads (in red the added sentences):

"The initial motivation of this work was studying the interplay between superconductivity and TER, using tunnel junctions composed of two superconducting electrodes that sandwich a ferroelectric, and across which superconducting Cooper pairs and quasiparticles may contribute to the conduction. As we will discuss later, the junctions were designed to boost TER mechanisms related to ferroelectric switching that we described above^{10,16,19,20}. Unexpectedly, we found that these mechanisms are not dominant, and that effects qualitatively and quantitatively identical to the TER are produced instead by a reversible electrochemical (redox) reaction that leads to oxygen exchange between the junction electrodes. This mechanism produces a giant TER in junctions that have a ferroelectric barrier, and also in junctions that consist of two dissimilar electrodes placed in direct contact –i.e. with no third material, ferroelectric or other, placed in between them. Because the redox reaction affects the physical properties of the electrodes, this scenario is different from that of junctions (or capacitors) in which the resistance switching is not in the tunnelling regime and is dominated by electromigration-induced changes within the thick insulating material placed between the electrodes²⁴⁻²⁸."

Referee's comment # 2: “Regarding the new data and theoretical fits: the STEM EELS seems to show an **enhanced** oxygen concentration near the BFO on the YBCO side. Isn't this in contradiction to statement that that YBCO will naturally oxygen deplete at the interface during MoSi growth (top of page 5)?”

Response: The deoxygenation we referred to takes place over length scales of a few unit cells, while the effects noticed by the referee (if any) take place over a couple of atomic planes near the interface. Unfortunately, no meaningful conclusions can be raised in this respect. The STEM-EELS data exhibit too much noise to allow us to conclude on the variation of the O content near the interface atomic planes. Indeed, the OK single profile exhibits a tail which decays within a length scale $< 1\text{nm}$, which is at the interface region, but effects within such sub-nm lengths scales could very well be associated with artefacts such as surface amorphous layers. A sentence has been added in the text to highlight this uncertainty, but it does not affect our conclusions. Notice that beyond the first YBCO unit cell, when moving into the YBCO layer, the O *K* signal within the next cells drops significantly when compared to the BFO perovskite, pointing to the clear deoxygenation of the YBCO that was noted in the paper.

Changes made: In page 5, we added the sentence:

“While the noise in the data make it difficult to extract solid conclusions on the degree of oxygenation of the YBCO interface plane, one can see that the O content in the interfacial layers gets clearly reduced with respect to the BFO.”

Referee's comment # 3: *I am still confused about the similar values for the switching fields in the very different cases (no BFO and thick BFO) – surely the voltage drop across the heterostructures mesa is very different in the case when a highly resistive barrier is used, and one might expect the electrochemistry to occur at quite different voltages. Of course there is the caveat of the fringing fields around the perimeter being important, but still I think my argument holds. In this context I don't understand the comment in section 1 of the supplementary “The voltage required to reverse... is dominated by the large E_0 for YBCO”, and a related statement in the main text at the top of page 10.”*

Response: We understand that this comment may be confusing.

As the referee states, the switching voltages are roughly the same regardless of the presence and nature and thickness of the interlayer between the electrodes. This is a clear experimental fact. As discussed in the paper, the natural explanation is that the system behaves similar to a “battery”, and therefore in a first approximation, what determines the voltage in the reduction potential of the electrodes. Thus, the voltage necessary for “charging” the battery (switch from OFF –ground state– into ON) is essentially given by the difference between the reduction potentials of the electrodes ΔE_0 . This allows for a quantitative explanation of the switching voltages, and also of the fact that they are essentially independent of the interlayer material. Since the reduction potential E_0 for YBCO is significantly larger than those of the different electrodes (around 0.2-0.4 V in absolute value), ΔE_0 is dominated by strong tendency of YBCO to be reduced. It is in this sense that we stated that the switching field is dominated by the large E_0 of YBCO.

Changes made: The comment “*is dominated by the large E_0 for YBCO*”, which is superficial in the context of the discussion on ΔE_0 in the SI, has been removed to avoid confusion.

Referee's comment # 4: “On page 11 they state that the barrier for ion transport arises at the YBCO interfacial layer. Is this obvious? Why is it not at the MoSi interface?”

Response: The switching from the ON into the OFF (ground) state involves a redox reaction that is spontaneous from the electrochemical point of view. Thus, the only energy barrier to be overcome is the barrier for ion transport. Consistently, we observe that the switching voltage from the ON into the OFF state is much lower ($V \sim -1$ V) than for the OFF into ON switching. Furthermore, we observe that the voltage is independent on the presence and nature of the interlayer, and also on the nature top electrode (MoSi or ITO behave very similarly as shown in the Supplementary Information). This suggests that the barrier for ion motion arises at the YBCO electrode, since it is the only material common to all the studied junctions. If the barrier arose from the MoSi interface, then the barrier for ion motion should be coincidentally similar in MoSi and ITO. This cannot be excluded, but it seems a stronger assumption.

Changes made: In Page 11, we have rephrased the statement to recall the fact that the switching field does not depend on the top electrode, and rephrased the statement to make it less categorical

“The fact that all junctions show the same ON to OFF switching $V_{\text{pol}} \sim -1$ V, regardless of the presence of an interlayer and of the top electrode material (see Fig. SI.5), suggests that the barrier for ion transport arises at the interfacial YBCO layer.”

Referee's comment # 5: *The BTK fits are poor, and I do not have confidence in the values extracted from these fits (for a gap or transparency).*

Changes made: Following the literature, we have improved the fits to the BTK model by considering the presence of inelastic scattering, a spatial inhomogeneity in the energy-gap, as well as by removing the bias dependence of the background conductance (evident for $eV > \Delta$) that is not accounted for by the BTK model. These precautions have allowed us to obtain better fits and to extract more reliable the parameters in the ON and OFF states. The fits are detailed in the supplementary information (section 7), and the key information obtained from these is explicitly referred in the main text.

Referee's comment # 6: “Similarly the discussion of the BDR fit was very brief (the equation fitted too, with any assumptions, was not shown, and the errors on the extracted parameters were not stated). It would be nice if these values could be referred to in the main text, and used to build a band alignment picture of these devices. These Supplementary data are measured at what temperature? There is a lack of information in many of the figure captions in this regard.”

Response: Thank you for pointing the lack of information. We have amended this section in the Supplementary following the suggestions. We have included a profile of the conduction band edge obtained from the BDR fit of the barrier height. As described below, the increase in barrier height observed in samples with BFO is due to its lower work function as compared to YBCO.

Changes made:

We have provided further details on the BDR fit, and the key information obtained is now explicitly referred to in the main text:

“Further quantitative analysis is shown in the Supplementary Information (see Figs. SI.6 and SI.7), including fits of the normal-state tunnel conductance allowing for estimates of the effective tunnel barrier thickness d and height $\bar{\varphi}$. Those yield $d \sim 5$ nm and $\bar{\varphi} \sim 0.15$ eV for junctions with 0 nm BFO and, consistently, $d \sim 8$ nm and $\bar{\varphi} \sim 0.53$ eV for junctions with 3 nm BFO. In absence of BFO, the 0.15 eV barrier is due to the decrease in YBCO work function resulting from the electron doping by the oxygen vacancies. The average barrier height $\bar{\varphi} \sim 0.53$ eV in samples with BFO barrier is explained by the difference of the work functions of electron doped BFO (4.7 eV) and YBCO (5.2 eV)”

We include a sketch of the tunnel barrier (Fig. SI.7) to better visualize the fitting results.

Referee's comment # 6: “Minor details:

-The statement that the Fermi screening length is expectedly **shorter** in the YBCO compared to the MoSi should be **longer**. T

-The authors should state their lower measurement limit of conductivity (“unmeasurably low”, on page 5).

Figure 1: could the authors put some indication on the TEM cross-section of the location of the chemical interfaces (as they have done in the integrated intensity vs depth plot using grey dashed lines).”

Changes: We have done all of the above corrections/changes.

Reviewers' Comments:

Reviewer #2:

Remarks to the Author:

The authors have answered my final questions. However I find it very awkward that the results of the quantitative analysis are only given in the graphs in the supplementary. I see a quantitative model, of the main experimental findings, as part of the main manuscript.

Reviewer #5:

Remarks to the Author:

I have read the manuscript and rebuttal letter. In my opinion the Authors have answered to all the Referees remarks.

I recommend publication.

There are still few typos to be corrected.

Response to referee # 2

Referee's comment 1 *“The authors have answered my final questions. However I find it very awkward that the results of the quantitative analysis are only given in the graphs in the supplementary. I see a quantitative model, of the main experimental findings, as part of the main manuscript.”*

Response : We thank the referee for his/her positive report. Following his/her opinion, we have now moved the results of the quantitative model to the main manuscript (which includes the new Figure 5), leaving only the technicalities (fitting procedure) in the Supplementary Information.

Response to referee # 5

Referee's comment 1 *“I have read the manuscript and rebuttal letter. In my opinion the Authors have answered to all the Referees remarks. I recommend publication. There are still few typos to be corrected.”*

Response : We thank the referee for his/her positive report. We have revised the manuscript and made our best to correct all the typos.